# C-Fiber Assays in the Cornea vs. Skin

**DOI:** 10.3390/brainsci9110320

**Published:** 2019-11-12

**Authors:** Eric A. Moulton, David Borsook

**Affiliations:** 1Center for Pain and the Brain, Department of Anesthesiology, Critical Care and Pain Medicine, Boston Children’s Hospital, Harvard Medical School, 1 Autumn St, Boston, MA 02115, USA; david.borsook@childrens.harvard.edu; 2Department of Ophthalmology, Boston Children’s Hospital, Harvard Medical School, 300 Longwood Ave, Boston, MA 02115, USA

**Keywords:** in vivo corneal confocal microscopy, corneal nerves, neuropathic pain, innervation, small fiber neuropathy, subbasal nerve plexus, nociceptors, intraepidermal skin biopsy

## Abstract

C-fibers are unmyelinated nerve fibers that transmit high threshold mechanical, thermal, and chemical signals that are associated with pain sensations. This review examines current literature on measuring altered peripheral nerve morphology and discusses the most relevant aspects of corneal microscopy, especially whether corneal imaging presents significant method advantages over skin biopsy. Given its relative merits, corneal confocal microscopy would seem to be a more practical and patient-centric approach than utilizing skin biopsies.

## 1. Introduction

The morphological properties of small sensory nerve fibers provide important markers of overall health of the peripheral nervous system. For example, a diagnosis of small fiber neuropathy (SFN) requires evaluating a skin biopsy for small fiber afferent health. Though invasive and presenting a risk of bleeding and infection, skin biopsy remains the clinical gold standard to assess the health of these afferents. With the advent of laser scanning in vivo confocal microscopy performed in the human cornea, small-fiber afferents can now be imaged noninvasively and at high spatial resolution. This paper will focus on C-fibers and briefly summarize the literature regarding anatomy of corneal innervation, present the way corneal afferent imaging may be used as a tool in the study of sensation and pain, and discuss potential benefits and utility of corneal microscopy relative to traditional skin biopsy.

The cornea is the most densely innervated structure in mammals and has been reviewed in detail elsewhere [1]. It provides a number of unique features for clinical examination because it can be scanned in awake human subjects, has a well-defined anatomy in health, and can exhibit changes in both neural integrity and inflammatory cells in patients. Afferents within the cornea consist of C- and A-delta fibers (70% vs. 30% by number, in the mouse) [2], including polymodal nociceptors (70%), mechano-nociceptors (20%), and cold thermoreceptors (10%). Derived from the long ciliary nerves, which extend from the nasociliary branch of the ophthalmic division of the trigeminal nerve (Cranial Nerve V), these sensory nerve fibers enter the globe medial and lateral to the optic nerve, course through in the suprachoroidal space, and branch to form nerve bundles that encircle the corneoscleral limbus and make up the limbal plexus. From the plexus, nerve trunks enter the corneal stroma radially and then ascend to innervate the corneal epithelium as free nerve endings through the subbasal nerve plexus, which primarily consists of C-fibers. As shown in the periphery [3], silent nociceptors are present and activated when the milieu is inflamed [4]. Unlike somatic nerve innervation, the cornea lacks A-beta fibers, and fibers involved in autonomic function are sparsely present [5]. Changes in corneal nerve function and structure through direct damage, metabolic changes, or systemic inflammatory processes may contribute to changes in corneal morphology.

In chronic pain conditions, nerve morphology alterations in the skin and cornea have been correlated with disease condition in both the peripheral and central nervous system [6,7,8]. However, these skin biopsy findings are not universal for all neuropathic pain conditions [9,10]. While skin biopsies have been sensitive for many small fiber or mixed neuropathies, the sensitivity and specificity of corneal nerve evaluation using corneal confocal microscopy (CCM) is less well defined. Furthermore, differences or similarities between these methods have not been stringently evaluated. While differences in fiber density across different body sites may contribute to variable sensitivity to stimuli, the overriding questions are: If small fibers are affected by a disease, is there also widespread fiber change?; can disease conditions result in alterations in small-fiber density in both the cornea and skin?

Quantitative approaches to measuring altered nerve morphology may contribute to understanding a disease state or its responsiveness to treatment. Some 7000+ PubMed citations are listed for the search term “skin biopsy and pain” and some 22 for “corneal nerve measures and pain”. While the overall sense is that both are sensitive to alterations in innervation, the issue is whether corneal imaging presents significant advantages over skin biopsy. Skin biopsies can evaluate patients with symptoms of numbness, tingling, or pain. Corneal imaging can evaluate patients with symptoms of itching, pain, discomfort, photophobia, and intolerance to cold. Both can be used to assess the health of small fibers in systemic disorders [11,12].

## 2. C-Fibers—Morphological Dynamics

In the context of baseline measures and responses to treatments, C-fiber health must be considered. C-fiber stimulation is associated with evoking a number of different sensations, including pain, warmth, itching, and sensual touch. Furthermore, a subclass of C-fibers demonstrate hyper-responsivity in diabetic neuropathy [13]. Sensitized C-fibers are more responsive to suprathreshold mechanical stimuli vs. uninjured control fibers [14]. With respect to functional properties associated with C-fiber nerve fiber density, physical regeneration and sensitivity of regenerated fibers are important to consider. Peripheral afferents can regenerate up to approximately 3.2 mm per day after unconditioned nerve crush injuries in a rodent model [15], and regeneration rates in disease states may contribute to opportunities for understanding drug efficacy in disease modification. Insight into this process in humans has been evaluated using a capsaicin-induced denervation model, which resulted in nearly complete degeneration of epidermal and subepidermal nerves [16]. In healthy volunteers, this intradermal capsaicin denervation model demonstrated reinnervation and recovery of limited sensation just three weeks post-injection. Compared with healthy subjects, the rate of nerve fiber density regeneration is significantly reduced in patients with diabetes [17], as well as in Human Immunodeficiency Virus (HIV) infection [18]. Importantly, regeneration may be affected by a number of factors, including mitochondrial dynamics within the nerve cell bodies [19], as well as presence of cytokines/macrophages [20]. Thus, measures of nerve fiber regeneration in disease states, or in response to a potentially modifying treatment, must account for issues related to the structural state of C-fibers, their rate of regeneration, and their subsequent altered functional sensitivity.

## 3. Methods for Quantitation of C-Fiber Changes

### 3.1. Intraepidermal Skin Biopsy (IESB)

Skin biopsy can be used for measurements of intraepidermal nerve fiber density (IENFD) of C-fibers (unmyelinated nerves) [21]. Using protein gene product (PGP) antibody labeling, which binds to unmyelinated nerve fibers, the density of these nociceptive fibers can be evaluated. Intraepidermal skin biopsy (IESB) is widely available and provides an objective assay of changes in distal nerve fiber integrity:
Data Collection: IESB is a relatively simple in approach, can be performed in clinical outpatient offices, and has commercial kits available for the procedure. Punch biopsies of the skin are taken from the region of interest, and the linear density of IENFD is quantified. The procedure, as defined previously, includes examining “at least three 50-µm-thick sections per biopsy, fixed in 2% paraformaldehyde lysine periodate or Zamboni’s solution, by bright-field immunohistochemistry or immunofluorescence, with anti-PGP 9.5 antibodies” [22]. The antibody PGP9.5 (protein gene product 9.5) is a neuron-specific protein that allows labeling of neurons and nerve fibers at all levels of the nervous system. Currently, a standard battery of antibodies to differentiate nerve fibers, such as C vs. A-delta vs. A-beta, has not yet been established. Such a test would be helpful for differential diagnoses of pathologies that differentially affect these nerve fibers, such as SFN.Data Analysis: Analysis involves quantification of fibers that intersect the dermal–epidermal basement membrane. A number of laboratories can now provide quantification of the skin biopsy. Details are provided elsewhere [21].Sensitivity and Specificity: IESB is considered to be reliable and reproducible [23]. Although the state or stage of the disease is difficult to define, multiple skin biopsies may be taken, allowing for longitudinal measurement of relative changes in a particular patient, as well as evaluation of clinical correlations, including disease etiology. Reduced IENFD is both sensitive and specific for well-defined clinical syndromes, but clinical phenotyping can be nonspecific. For example, in a retrospective report, IENFD abnormalities were detected in 88% of patients with symptoms suggestive of SFN, which was significantly better compared with a clinical exam or quantitative sensory testing (QST) [24]. In addition, IENFD was abnormal in approximately 80% of clinically defined mixed large- and small-fiber neuropathies, with no changes observed in large-fiber neuropathy.Disease Evaluation: Skin biopsy for small-fiber abnormalities has been shown across well-described conditions that have SFN (e.g., HIV, diabetes, chemotherapy-induced neuropathy), mixed peripheral neuropathies, and also in conditions having less defined pathophysiologies (e.g., Parkinson’s or fibromyalgia) and autonomic neuropathies. The European Federation of Neurological Societies has provided guidelines for the use of skin biopsy in the diagnosis of peripheral neuropathies [22]. Diagnostic specificity and sensitivity using IESB is high. Normative data has been provided in healthy controls and neuropathic patients [25]. These authors reported the number of intraepidermal fibers in normal controls to be 21.1 ± 10.4 per mm (mean ± SD) in the thigh (fifth percentile, 5.2 per mm) and 13.8 ± 6.7 per mm at the distal part of the leg (fifth percentile, 3.8 per mm). Using a cutoff taken from the fifth percentile of healthy data, they concluded that the approach had “a positive predictive value of 75%, a negative predictive value of 90%, and a diagnostic efficiency of 88%”.Data interpretation: IESB has been useful in evaluating small-fiber neuropathies across multiple small-fiber neuropathies and mixed-fiber neuropathies (see Table 1). IEDNF is reduced in patients with small-fiber neuropathy and mixed-fiber neuropathy, but not in large-fiber neuropathies. Because C-fibers function for thermal and pain transduction, thermal testing for noxious heat can be a useful or confirmatory adjunct [16].

#### 3.1.1. State Evaluation

(1)Static evaluation: Single nerve biopsies may be useful in diagnosis, but evaluation across gender and race/ethnicity has not been fully researched. For example, skin biopsy values in a Chinese population for diagnosing SFN have shown low sensitivity [86].(2)Dynamic evaluation: Despite the capacity for repeated measurements, few longitudinal studies have evaluated the utility of IESB. Some studies have evaluated the effects of age on epidermal sensory innervation [87]. Their findings noted an age-associated decrease in the facial biopsies, but not in biopsies derived from the abdomen. Evaluation of SFN in three clinical groups (idiopathic, impaired glucose tolerance, and diabetes) showed similar decreases over three time points at three sites (proximal thigh, distal thigh, and distal leg), noting similar rates of decrease in the different clinical categories over time [7]. Thus, axon loss may not be length-dependent over time, and the process inducing these changes may be local rather than at the level of the cell bodies [88]. However, IESB measures of SFN have been associated with central nervous system changes in patients following the development of central post-stroke pain [6]. It was unclear whether there was a true correlation/causality of the disease or whether the age of the patients in central post-stroke pain played a role (all were >50 years old), although none of the patients had a diagnosis consistent with coexistent SFN and the extent of reduced fiber density observed in the calf (<3.4 fibers per mm) was well below normal, even when accounting for age. Of note, a study in Parkinson’s patients detected no changes in IEDNF, although decreases in corneal nerve-fiber density and length were detected using CCM [89].

#### 3.1.2. Autonomic Nerve Assessment [21]

A complex relationship exists between epidermal fibers and autonomic function. Changes in IENFD have been observed in conditions, such as amyotrophic lateral sclerosis (ALS)—not previously thought to have a pathophysiology affecting epidermal nerves [90]. In ALS, one mechanism of alteration in nerve fibers may relate to the deposition of transactive response DNA-binding protein (TDP)-43/pTDP-43 [91]

### 3.2. Corneal Confocal Microscopy (CCM)

CCM is a method for evaluating alterations in corneal fibers in vivo using confocal microscopy. Unlike skin biopsy, the approach is noninvasive, easily repeated for the same region, and offers an alternative method to detect changes in nerve morphology for various pain conditions, both peripheral and central (see Table 1). Though imperfect (see limitations below), measures of altered corneal nerve morphology may be considered a marker for nerve damage [92]. CCM provides more information on nerve morphometry than can be obtained by skin biopsies, including fiber length and fiber branching. The association between abnormal corneal nerve morphology and generalized pain conditions (e.g., fibromyalgia) is not well understood. The ability to differentiate C vs. A-delta fibers in the cornea is limited as myelination, a defining morphological feature of A-delta fibers, does not extend significantly into the cornea.
Data collection: CCM is an outpatient procedure that requires specialized equipment, such as a Heidelberg Retinal Tomograph II/III with a Rostock Corneal Module. Because the microscope lens makes brief contact with the cornea, an anesthetizing eye drop of 0.5% proparacaine is applied prior to image acquisition. The procedure usually takes less than 15 min. Data is electronically captured and stored for analysis. Approach and data collection have been described in detail elsewhere [27].Data analysis: Analysis may be performed manually or, more recently, using a fully automated approach [93,94,95]. Using these approaches, data may be collected in the categories of: Corneal nerve fiber density (CNFD), branch density (CNBD), and nerve fiber length (CNFL) and width (CNFW). For example, CNFD is a measure of the major corneal nerves and is expressed in fibers per mm^2^; CNBD measures the density of nerve branches off the major corneal nerves using the number of branch points on main fibers per mm^2^; CNFL reflects the total length of the nerves in mm per mm^2^. Taken together, these measurements can be used for evaluating disease state (i.e., severity, progression, regression) and treatment effects for diseases affecting the C-fibers. Normative data is also available to facilitate comparisons [27].Sensitivity and Specificity: CCM is a sensitive method to evaluate corneal metrics in healthy and disease states. Methods for improving sensitivity and specificity include sampling an average of three representative standard images [96], as well as large-area mosaic imaging of the subbasal nerve plexus [97]. Automated vs. manual evaluation of corneal fiber metrics show consistency within and across evaluators [98]. Comparisons of CCM and IESB measures have shown an equivocal sensitivity and specificity in the detection of diabetic sensorimotor polyneuropathy [99] and sometimes higher sensitivity with CCM. In a comparative study in the detection of diabetic neuropathy, CNFD showed slightly higher diagnostic sensitivity than IESB (0.77 vs. 0.61) and comparable specificity (0.79 vs. 0.80) [52]. Another comparative study, this one in the detection of sarcoidosis, found that IESB was again less sensitive than CCM, with decreased IENFD found in 28% of patients, while CCM abnormalities were present in 45% of patients [48]. However, studies need to be performed to further understand the correlation between these two approaches, for their utility for diabetes and sarcoidosis, as well as other disease states.Disease Evaluation: CCM offers the opportunity to (a) define changes in small fibers in a disease ahead of clinical symptomatology; (b) potentially predict disease onset [100]; and (c) possibly define treatment efficacy in an objective manner [101,102]. Changes in corneal innervation may be evaluated for specific corneal pathology and for systemic disease. While changes in corneal innervation are not specific to pain, most pain conditions studied to date show altered CCM metrics (see Table 1). However, other diseases, such as multiple sclerosis or Parkinson’s disease, also show changes. Since pain can be associated with these diseases, changes may represent the development of pre-pain conditions; for example, Parkinson’s disease is associated with central and peripheral neuropathic or muscular pain.Data Interpretation: Changes in nerve integrity may correlate with systemic disease, including pain, but may also be altered in other neurological conditions, such as multiple sclerosis [103], Parkinson’s disease [68], Friedreich’s ataxia [76], or chronic demyelinating polyneuropathy [104]. Importantly, changes may reflect systemic rather than local symptomatology, as has been shown with a prediction of the development of peripheral neuropathy in diabetic neuropathy [105]. Only a few studies have indicated that CCM can be utilized to evaluate a therapeutic response to treatment [101]. A few corneal indices, such as investigation of the inferior whorl, may also help define painful vs. non-painful states, although this observation needs to be studied further in pain vs. non-pain states [106].
(1)State evaluation: Similar to IESB, CCM measures across sex and race/ethnicity are still an active area of investigation. For example, a meta-analysis of normative values in a 343-healthy-subject cohort did not show sex differences in CCM measures [27].(2)Dynamic evaluation: While repeated IEDNF measures are possible with skin biopsies adjacent to each other, CCM allows for multiple measures across time in the same sampling region. This allows for dynamic evaluation of afferent health in response to condition or therapy.

## 4. Skin Biopsy vs. Corneal Microscopy 

### 4.1. Age Domain

CCM may be conducted across ages, at least from 8 years or older. No injections of local anesthetic or “traumatic” injury are required. If standard precautions for infection prevention are not followed, there is a risk of ocular infection. While a theoretical risk exists with excessive light exposure, photochemical damage has been estimated to occur when continuous exposure exceeds 8 h with optical coherence tomography, the technological basis of CCM [107]. Skin biopsies can be performed at any age, though special considerations are necessary as it is a minor invasive procedure [108].

### 4.2. Disease Domain

Changes are observed in both skin and corneal measures in “systemic” diseases (Table 1, Table 2), such as Parkinson’s disease. IENFD (including myelinated nerves innervating Meissner corpuscles) is reported as lower in Parkinson’s disease patients vs. controls, with a greater impact on the body side with greater symptoms [109]. CCM has shown significant decreases in CNFD, CNBD, and CNFL in patients with dementia and even mild cognitive impairment [110]. These findings highlight the equal utility of each technique to potentially monitor systemic changes in the peripheral and central nervous system (Table 1).

### 4.3. Temporal Domain

CCM can be evaluated longitudinally, measuring within the same area. To reduce variability due to sampling area selection, automated spatial registration of the cornea remains an area in active development. The amount of time required for some corneal measures to return to normal, including ongoing regeneration, may take years [111].

### 4.4. Degeneration and Regeneration

Degeneration in corneal nerve morphology is evidenced by multiple changes, including nerve tortuosity, nerve fiber orientation, and decreased nerve density. For local conditions, such as corneal nerve damage following laser-assisted in situ keratomileusis (LASIK) [111,112], CCM evaluation can provide an objective assay of recovery that may correlate with clinical symptoms. A number of studies have used CCM to evaluate corneal nerve regeneration [113]. IESB has also been used to study regeneration, as abnormalities in nerve regeneration rates have also been demonstrated in patients with HIV [18,63] and diabetes [17]. We are unaware of any study that has directly correlated corneal nerve regeneration with pain levels or pain reversal, and IESB measures do not correlate well with neuropathic pain with peripheral neuropathies [9]. Corneal nerve migration rate is a useful measure of regeneration; in healthy subjects it has been reported to be approximately 42 µm per week with no sex or age differences [114]. Neurotrophic factors have been identified that promote nerve regeneration in animal models [115,116,117] as well as humans [118].

### 4.5. Correlative Measures

Other methods of evaluation of C-fiber loss or dysfunction in the skin include quantitative sensory testing (QST) [119] and laser doppler imaging flare [120]. For example, thermal sensory testing provides a behavioral response consistent with C-fiber loss and can readily evaluate skin regions across diseases [121]. In the case of the cornea, menthol produces uncomfortable sensations in patients with dry eye, a potentially neuropathic condition [122]. Although cold sensitivity is considered to be part of corneal fiber loss or changes (e.g., dry eye) and is also associated with neuropathic pain conditions [123,124], corneal sensitivity tests, also known as corneal esthesiometry, include the Belmonte and Cochet–Bonnet aesthesiometers and the relatively crude corneal reflex test using a cotton swab. C-fiber structure in the cornea is not routinely evaluated across pain conditions at this time.

### 4.6. Quantitation Domain

Although IEDNF is considered a reproducible and reliable method [23], this is controversial [125]. Evaluation requires preparation via staining, and quantitation is not automated; it has been suggested that reproducibility is higher if three images acquired from different areas of the cornea are analyzed [23]. CCM provides essentially immediate results and can be quantitated in minutes. The current evaluative methods seem to be reproducible within and across observers and with manual vs. automated counting by at least two separate research groups [98,126,127]. For clinical diagnostic testing, manual evaluation of the needed number of sections is more tedious and less cost-effective than automated analysis. Cross-study reproducibility is key to evolving CCM to clinical utility and is an important area to establish with future studies.

### 4.7. Predictive Domain

Measures of changes in nerve fiber metrics can be more easily evaluated using the CCM approach, because both local and systemic effects can be evaluated. CCM changes have predicted changes in the future onset of neuropathy disease severity in diabetes type 1 [128] but are not well defined in other disease domains, nor in the context of pain. Specific measures, such as nerve fiber length, may be more sensitive in differentiating diabetics from healthy patients and thus may have some predictive value as a screening test [100,105,129].

### 4.8. Clinical Models of Nerve Changes

Some clinical treatments provide ideal models with which to evaluate changes in fiber density and reinnervation. Examples include: (1) Nerve damage treatments, including post-LASIK—a procedure that severs corneal nerves [130,131]; (2) toxic treatments, such as chemotherapy [41], in which CCM has the potential to be more sensitive than IESB [10]; and (3) endocrine-related treatments in diabetic neuropathy. For direct trauma/surgery to the skin, IESB has not been used in humans but has been observed in entrapment neuropathy in rats, a model for chronic compression injury [132].

### 4.9. Drug Evaluation

Open access to the cornea is a good reason to utilize measures of nerve morphology as an index of drug efficacy and disease modification. Since changes in the cornea may predict disease onset (e.g., diabetic peripheral neuropathy), the opposite may also be true. Both systemic and local effects of drugs on corneal nerves can be evaluated, whether to assess therapeutic efficacy or as a pharmaceutical side effect. A robust approach to this—considering dosing and time for effect—has not yet been conducted. However, supportive data for such an approach is a compelling need.

## 5. Conclusions

Evaluation of nerve health provides insight into dynamic changes of the peripheral nervous system during disease states or treatment effects. The notion of a “peripheral-only” or “central-only” pathogenesis of pain seems to be progressively eroding as our understanding of the interaction between peripheral and central nervous system changes takes place in pain conditions. The cornea provides an ideal window into these processes. While more information is required to define corneal changes as true biomarkers, it would seem to be a more practical and patient-centric approach than utilizing skin biopsies.

## Figures and Tables

**Table 1 brainsci-09-00320-t001:** Reports of cornea or skin-based nerve metrics on corneal confocal microscopy (CCM) or intraepidermal skin biopsy (IESB) measures in different conditions/disease states in humans.

Condition/Disease	Method	
	CCM	IESB
Physiological States		
Healthy	[26] 64 subjects—CNFL is not associated with age and contact lens wear [27] 343 subjects—height, weight, and body mass index do no influence 5th percentile normative values for any corneal nerve parameter	[28] 192 subjects—IENFD is associated with anatomic site, race, and age, and not gender
Aging	[29] 85 subjects—CNFD decreases with age [27] 343 subjects—decreased CNFD and CNFL with age; corneal nerve fiber tortuosity increases with age	[30] 10 subjects—little age-related change in IENFD [31] 528 subjects—IENFD is associated with age and gender
Capsaicin Application	-	[32] 10 subjects—IENFD decreased 3 days following capsaicin
Local Anesthetic Application	-	[33] 20 subjects—decreased IENFD after 42 days of lidocaine patch treatment
Eye-Related Conditions		
Dry Eye	[34] 24 non-Sjogren’s dry eye; 44 Sjogren’s dry eye; 24 controls— decreased CNFD, increased tortuosity, and increased DC in both dry eye groups	-
Laser-assisted in situ keratomileusis (LASIK)	[35] 3 patients—reduced CNFD	-
Herpes Zoster Ophthalmicus (HZO)	[36] 24 HZO affected eyes; 24 contralateral unaffected eyes; 24 controls—decreased CNFL, CNFD, CNFB, and increased DC density	-
Pain-Related Conditions		
Chronic Migraine	[37] 19 patients; 30 controls—decreased CNFD [38] 24 migraineurs w/photophobia; 24 migraineurs w/o photophobia; 24 controls—decreased CNFL, CNFA, CTBD, CNBD	-
Fibromyalgia	[39] 34 patients; 42 controls—decreased total nerve density, long nerve fibers, number of nerves [40] 17 patients; controls—decreased CNFD	[8] 46 patients; 34 controls—reduced IENFD
Chemotherapy-Induced Neuropathy	[41] 21 patients; 21 controls—decreased CNFD, CNBD, and CNFL; increased CNFL after 3rd cycle of chemotherapy	[10] 13 patients; 47 controls—no significant decrease in IENFD
Herpes Zoster Ophthalmicus	[36] 24 affected eyes; 24 unaffected contralateral eyes: 24 controls—increased DC in both eyes	-
Small-Fiber Neuropathy (SFN)	[42] 14 patients; 14 controls—decreased CNFD and total number of nerves; increased nerve tortuosity. [40] 17 patients; 17 patients—decreased CNFD	[7] 25 idiopathic SFN; 13 impaired glucose tolerance (IGT)-associated SFN; 14 diabetes mellitus (DM)-associated SFN—decreased IENFD with IGT and DM vs. idiopathic SFN.
Pruritus	-	[43] review paper—decreased IENFD
Trigeminal Neuralgia (post-compression surgery of trigeminal ganglion)	[44] 21 affected eyes; 21 unaffected contralateral eyes—no change in CNFD	-
Immunological Disease		
Sjogren’s Disease	[34] 24 non-Sjogren’s dry eye; 44 Sjogren’s dry eye: 24 controls—decreased CNFD, increased nerve tortuosity, and DC with both dry eye groups [45] 30 patients; 12 controls—increased corneal nerve entropy	[46] 61 patients; 106 controls—decreased IENFD
Behcet’s Disease	[47] 49 patients; 30 controls—decreased CNFD and CNFL; increased DC density	-
Sarcoidosis	[48] 58 patients—decreased CNFL, CNFD, or CNBD in 45% of patients	[48] Decreased IENFD in 28% of patients [49] 72 patients; 188 controls—decreased IENFD
Systemic Lupus Erythematosus	[50] 27 patients; 27 controls Increased central DC density	[46] 60 patients; 106 controls—decreased IENFD [51] 60 patients—decreased IENFD
General Neuropathies		
Diabetic Neuropathy	[52] 30 Type 1 diabetes without neuropathy; 31 Type 1 diabetes with neuropathy; 27 controls—decreased CNFD, CNBD, and CNFL in patients with neuropathy vs. without and controls	[53] 32 recent-onset Type 2 diabetes; 34 diabetic sensorimotor polyneuropathy (DSPN) + pain; 32 DSPN-pain; 50 controls [54] 35 Type 2 diabetes; 32 metabolic syndromes; 36 metabolic syndromes for longitudinal testing
Metabolic Syndrome	-	[54] 35 Type 2 diabetes; 32 metabolic syndromes: 36 metabolic syndromes for longitudinal testing—decreased IENFD in both patient groups [55] 38 patients—decreased IENFD
Autonomic Neuropathy	[56] 3 patients; 3 controls—decreased CNFD	[21] review article—decreased IENFD
Vitamin D Deficiency	[57] case report—increased DC	-
Amyloid Polyneuropathy	[58] 15 patients; 15 controls—decreased CNFL	[59] 70 transthyretin (TTR)-mutant; 66 TTR-A975—decreased IENF density
Chronic Inflammatory Demyelinating Polyneuropathy (may include other dx)	[60] 16 patients; 15 controls—decreased CNFD and CNFL; increased nerve tortuosity	[46] 60 Systemic Lupus Erythematosus; 61 Sjogrens; 52 Rheumatoid Arthritis; 106 controls—decreased IENFD
Peripheral Autoimmune Neuropathy	[61] case report—increased tortuosity of stromal nerves; no change in subbasal nerves	-
Human Immunodeficiency Virus Neuropathy	[62] 20 patients; 20 controls—decreased CNFD, CNBD, and CNFL; increased tortuosity	[63] 62 patients [64] 101 distal leg, 99 proximal thighs—decreased IENFD
Arterial Ischemic Disease	-	[65] 20 symptomatic peripheral arterial disease; 20 critical leg ischemia; 12 controls—decreased IENFD
Central Nervous System (CNS) Disorders		
Charcot Marie Tooth	[66] 12 patients; 12 controls—decreased CNFD, CNBD, CNFL, and tortuosity	[67] 80 patients—decreased IENFD
Parkinson’s Disease	[68] 26 patients; 26 controls—decreased CNFD; increased CNBD and CNFL	[69] 28 patients; 23 controls—decreased IENFD [68] 24 patients; 10 controls—decreased IENFD
Stroke/Central Post-Stroke Pain	[70] 146 patients; 18 controls—decreased CNFD, CNBD, and CNFL	[6] 4 patients—decreased IENFD
Fabry Disease	[71] 10 heterozygous females w/Fabrys w/o enzyme replacement therapy (ERT); 6 heterozygous females w/Fabrys w/ERT; 6 heterozygous males w/Fabrys w/ERT; 14 controls—decreased CNFD and CNBD	[72] 120 patients—decreased IENFD
Amyotrophic Lateral Sclerosis (ALS)	-	[73] 8 ALS; 5 facial onset sensory and motor neuronopathy—decreased IENFD
Multiple Sclerosis	[74] 26 patients; 26 controls—decreased CNFD	-
Wilsons Disease	[75] 24 patients; 24 controls—decreased corneal nerve fiber density, number of branchings, number of beadings, and tortuosity.	-
Friedrich’s Ataxia	[76] 23 patients; 14 controls—decreased CNFD and CNFL	[77] 17 patients—decreased IENFD
Ehlers–Danlos Syndrome	[78] 50 patients; 50 controls—thinner stroma, lower keratocyte densities, increased applanation-related stromal folds, increased endothelial hyperreflective dots.	[79] 24 patients—decreased IENFD
Drug-Induced Conditions		
Pregabalin	-	[80] 26 subjects—IENFD lower in non-responders
Corticosteroids	[81] 50 subjects—decreased DC density	-
Cyclosporine	[82] 40 subjects—decreased CNFD, reflectivity, tortuosity, and hyperreflective keratocytes; increased cell density of intermediate epithelium cells	-
Cibenetide	[83] 64 subjects—increased CNFA	-
Crack Cocaine Use	[84] case report—decreased CNFD and CNFL; increased tortuosity	
Vitamin B Deficiency	-	[85] 10 patients; 10 controls—decreased IENFD

Corneal nerve branch density (CNBD), corneal nerve fiber area (CNFA), corneal nerve fiber branches (CNFB), corneal nerve fiber density (CNFD), corneal nerve fiber length (CNFL), corneal total branch density (CTBD), dendritic cells (DC), intraepidermal nerve fiber density (IENFD).

**Table 2 brainsci-09-00320-t002:** Relative advantages of CCM vs. IESB in evaluating C-fiber changes.

Condition	CCM	IESB
Age	Suitable for ages >8 years, noninvasive	All ages, but invasive; in some conditions, healing may be affected
Disease State	Many pain and non-pain conditions	Many pain and non-pain conditions
Temporal Measures	Temporal Dynamics easily evaluated, including short and long-term evaluation	Temporal dynamics not easily evaluated
Measures of Degeneration or Regeneration	Can evaluate the same region repeatedly	Can evaluate neighboring regions
Correlative Measures	Extraterritorial changes can be evaluated	Thermal pain using quantitative sensory testing
Quantitative Sensitivity	Good repeated measures across evaluators or automated measures	Good across evaluators assuming appropriate skin punch biopsy taken
Predictive Value for Neuropathy	Good	Good
Inflammatory Markers	Good; inflammatory cells can be visualized	Good; requires staining
Adaptive Clinical Models	Keratoplasty, to assess nerve fiber damage and regeneration	Capsaicin denervation, to assess nerve fiber retraction and reversal

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
