# Peer review of "C-Fiber Assays in the Cornea vs. Skin"

_brainsci, 2019, doi:10.3390/brainsci9110320_

Round 1

Reviewer 1 Report

C-Fiber Assays in the Cornea vs. Skin

Moulton and Borsook in Brainsci 615956

Overall, the review is very clearly, adeptly written and enjoyable to read, covering an important topic in the emerging field of corneal confocal microscopy as a diagnostic tool for neuropathies of various etiologies. There are a few points that may enhance the review.

Points of discussion:

In terms of the procedures, could the authors consider the following: Risks of the procedures: it is true that skin biopsies contain some risk and require local anesthetic but in reality, it is a minor invasive procedure. The authors seem to play up the risk here. The authors don’t mention the risk of ocular infection if the CCM is not properly disinfected (which also happens in clinics). Safety of the procedures: consider mentioning the safety data on light exposure to further describe the low risk of CCM. The authors could comment on the use of modified Belmonte or Cochet-Bonnet aesthesiometers or other corneal sensitivity tests to complete the comparison to the skin sensitivity tests described. The authors focus on C-fiber and not Ad projections, which is appropriate, but it would be helpful to perhaps include the following: Give the percentage of corneal nerve fibers that are C-fiber and Ad in the introduction. Discuss if CCM can differentially characterize the types of innervation by morphology or location. Comment on whether there is a good antibody test to identify Ad fibers in IESBs, and point out any potential for differential diagnoses. State upfront that this review will cover C-fibers (because of reasons 2a and 2b). I kept looking for comments on Ad projections, it would help readers stay focused. Not needed, but perhaps mention where the long ciliary and nasociliary nerves enter the globe in the introduction as a general “lay of the land”. Application to clinical conditions, could the authors consider the following: Mention damage from LASIK or cataract surgical earlier, it only comes up towards the end, and readers will be very aware of these. Increase clarity on which diseases/conditions show differences that can or could be used diagnostically (i.e. clear cutoff for pathology) versus for research. The 2 items below are examples of where it was not clear what is in common clinic practice versus development stage. Lines 218-219, comment of the altered corneal innervation in patients with dementia or mild cognitive impairment different as a measure with diagnostic potential. Lines 192-194 suggests the Parkinson’s disease and multiple sclerosis CCM results may represent development of a pre-pain condition. Equally well, the changes may be associated with other outcomes, which would also provide a great opportunity to look into pain-free, pre-pain and pain conditions. Data comparisons, could the authors consider the following: Some additional information would help Table 1, minimally the number of participants for each study but also possibly the major conclusion. The total number of participants in various CCM studies would give support for this becoming/being an established metric. Similarly, lines 173-185 would be helped with giving an idea of the “n” for the studies, even as a general comment to make the argument for the use CCM or the need for further comparative studies. Lines 137-139, the authors comment that a true causality could not be determined, can they comment on whether or not other studies with similarly healthy aged patients showed similar corneal innervation? If this is going to be recommended as a diagnostic tool, a comment on cross-study reproducibility is important, and if not at that capability yet, an important area to point out for further studies.

Minor points:

The abbreviation QST p3 line 104 is not explained, CCN is given instead of CCM in the heading and title of Table 1. Line 217: replace “more so for the more affected side” with something else as this was one of very few lines that was a bit confusing to read. Lines 225-226: could the authors specify if the “may take years” refers to the time for regeneration or technological and algorithmic considerations? Lines 247-249 are a little vague. Maybe say some also consider it controversial instead of “is somewhat controversial”. At the end “reproducibility is higher if four sections are counted” is vague. As written, it could be interpreted as 4 different biopsies or 4 sections from the same biopsy. Also, I think the authors point is that for clinical diagnostic testing, that evaluation of the needed number of sections is more tedious and less cost-effective than the automated analysis with CCM (pointed out elsewhere). The authors’ intent could be clarified here. Line 264, please give a very brief idea of what entrapment neuropathy in rats is (chronic compression injury) for the non-clinician.

Author Response

We would like to the reviewers for their thoughtful reading and suggestions for our paper.  We feel that their suggestions have made impactful improvements to our text.  We have incorporated their suggestions in the attached Word document (revisions viewable under Tracked Changes), which are summarized below point by point.

Reviewer #1

Points of discussion

State upfront that this review will cover C-fibers

This is an important point, and we have added the following to the first introductory paragraph:

Line 26 - This paper will focus on C-fibers, and briefly summarize the literature regarding anatomy of corneal innervation, present the way corneal afferent imaging may be used as a tool in the study of sensation and pain, and discuss potential benefits and utility of corneal microscopy relative to traditional skin biopsy.

Risk of procedures

Skin biopsies are a minor invasive procedure

            Mention risk of ocular infection if CCM not properly disinfected

We agree, and have added the following:

Line 384 - If standard precautions for infection prevention are not followed, there is a risk of ocular infection.

Line 387 - Skin biopsies can be performed at any age, though special considerations are necessary as it is a minor invasive procedure

Safety of procedures

Consider mentioning safety data on light exposure to describe low risk of CCM

We have added the following:

Line 384 - While a theoretical risk exists with excessive light exposure, photochemical damage has been estimated to occur when continuous exposure exceeds 8 hours with Optical Coherence Tomography, the technological basis of CCM

Comment on Belmonte or Cochet-Bonnet aesthesiometers or other corneal sensitivity tests to complete comparison to skin sensitivity tests described

Line 434 - Corneal sensitivity tests, also known as corneal esthesiometry, include the Belmonte and Cochet-Bonnet aesthesiometers, and the relatively crude corneal reflex test using a cotton swab.

Report percentage of corneal nerve fibers that are C-fiber and A-delta in introduction

Can CCM differentiate between Adelta and C by morphology or location

Comment on whether there is a good antibody test to identify Ad in IESB

point out potential for differential diagnoses

Line 34 - Afferents within the cornea consist of C and A-delta fibers (70% vs. 30% by number, in the mouse)

Line 97 - The antibody PGP9.5 (Protein gene product 9.5) is a neuron specific protein that allows labeling of neurons and nerve fibers at all levels of the nervous system. Currently, a standard battery of antibodies to differentiate nerve fibers, such as C vs. A-delta vs. A-beta, has not yet been established. Such a test would be helpful for differential diagnoses of pathologies that differentially affect these nerve fibers, such as SFN.

Options – mention where long ciliary and nasociliary nerves enter globe in intro layout

Line 35 - Derived from the long ciliary nerves, which extend from the nasociliary branch of the ophthalmic division of the trigeminal nerve (Cranial Nerve V), these sensory nerve fibers enter the globe medial and lateral to the optic nerve, course through in the suprachoroidal space, and branch to form nerve bundles that encircle the corneoscleral limbus and make up the limbal plexus.

Application to clinical conditions

Mention damage from LASIK or cataract surgical earlier (currently late)

We have moved mention of LASIK earlier in two different areas within the text:

Line 416 - Degeneration and Regeneration: Degeneration in corneal nerve morphology is evidenced by multiple changes, including nerve tortuosity, nerve fiber orientation, and decreased nerve density. For local conditions such as corneal nerve damage following LASIK, CCM evaluation can provide an objective assay of recovery that may correlate with clinical symptoms.

Line 453 - Clinical Models of Nerve Changes: Some clinical treatments provide ideal models with which to evaluate changes in fiber density and reinnervation. Examples include: (1) nerve damage treatments, including post-LASIK—a procedure that severs corneal nerves

Increase clarity on which diseases/conditions show differences that can/could be used diagnostically (clear cutoff for pathology) vs. research E.g. Parkinson’s and MS CCM results(lines 192-194)

We believe that to include clinical utility judgments on the multitudes of varied conditions presented goes beyond the intended scope of our paper. We would like to explore this in a future separate publication.

Data comparisons

Table 1 – patient number – major conclusion

Lines 173-185 – include n for studies

For Table 1, we have added subject numbers and major conclusions for each paper as appropriate.

Lines 137-139 – do other studies with similarly healthy aged patients show similar

corneal innervation

Line 147 - It was unclear whether there was a true correlation/causality of the disease or whether the age of the patients in CPSP played a role (all were >50 years old), although none of the patients had a diagnosis consistent with coexistent SFN and the extent of reduced fiber density observed in the calf (<3.4 fibers per mm) was well below normal even when accounting for age.

Comment on importance of cross-study reproducibility

If not at that capability yet, an important area to point out for further studies

Line 445 - Cross-study reproducibility is key to evolving CCM to clinical utility, and is an important area to establish with future studies.

All of the below minor points have been incorporated into the text as suggested (see Tracked Changes in the text for details).

Minor points

Define QST – p3, line 104

CCN instead of CCM in Table 1 heading and title

Line 217 – replace “more so for the more affected side”

Line 225-226 – “may take years” refers to time for regeneration or tech and alg consideration

Line 247-249 – vague

“is somewhat controversial” -> controversial

“Reproducibility is higher if four sections are counted” is vague

            could be interpreted as 4 different biopsies or 4 sections from same biopsy

Clarify that for clinical diagnostic testing, evaluation of the needed number of sections is more tedious and less cost-effective than the automated analysis with CCM

Line 264 – describe entrapment neuropathy in rats (chronic compression injury)

Reviewer 2 Report

The manuscript titled as "mC-Fiber Assays in the Cornea vs. Skin" is a review describing altered peripheral nerve morphology and various important aspects of corneal microscopy. The literature reviews that corneal imaging is superior to skin biopsy.The manuscript is well written covering all important aspects of methods, and literature. However there are some minor English language errors and the manuscript needs to be proof read to correct them.

Author Response

We would like to the reviewers for their thoughtful reading and suggestions for our paper.  We feel that their suggestions have made impactful improvements to our text.  We have incorporated their suggestions in the attached Word document (revisions viewable under Tracked Changes), which are summarized below point by point.

Reviewer #2

There are some minor English language errors and the manuscript needs to be proof read to correct them.

The manuscript has been reviewed by our departmental science writer.